# Evaluating Emergency Remote Assessment Adaptations in Higher Education due to COVID-19: Faculty Insights and Challenges

Elena C. Papanastasiou [1,*] and Georgia Solomonidou [2,*]

1   School of Education, University of Nicosia, Nicosia 2417, Cyprus
2   School of Humanities, Social and Education Sciences, European University Cyprus, Nicosia 2404, Cyprus
*   Correspondence: papanastasiou.e@unic.ac.cy (E.C.P.); g.solomonidou@research.euc.ac.cy (G.S.)

**Abstract:** The purpose of this study was to critically examine the feedback obtained from higher education instructors regarding the implementation of the emergency remote assessment practices that took place within a university in the Republic of Cyprus, in order to identify the strengths and weaknesses of the changes that took place. This was essential since the abruptness of the pandemic did not always allow for smooth transitions during the introduction of these changes. Therefore, the results of this survey study that was based on an online questionnaire identified certain aspects of the assessment adaptations that were evaluated as positive (e.g., the use of e-invigilation software), and other aspects that were not as positive (e.g., performing oral examinations after the written test). However, the results also revealed that cheating and plagiarism were issues that concerned the instructors, as were the technological problems that were faced. All these results are discussed holistically at the end of this article in order to guide further research and decision making regarding online assessments.

**Keywords:** online testing; COVID-19 pandemic; tertiary education; test adaptations; oral assessment; cheating; plagiarism software; e-invigilation; final examinations

## 1. Introduction

While going through the third year of the COVID-19 pandemic, it has become all too evident throughout the globe that no aspect of human life has remained unaffected. Beyond its known effects on health, the pandemic has had a major impact in the field of education. Numbers speak for themselves as, according to the Organization of Economic Cooperation and Development (OECD) [1], in 2020, 1.5 billion students in 188 countries were locked out of schools of all levels. Students and teachers universally had to deal with uncertainty of their next steps whilst schools were open one day and closed the next, causing obviously substantial disruption to teaching and learning. Each educational level and discipline faced its own unique challenges [2–4]. In the field of higher education (HE), one of the main challenges that was faced universally was the need to make an emergency transition to online teaching and online examinations [5–9].

Although adapting teaching and assessment to an online environment during crises is not new, the current situation was characterized as unique as it has not affected only a specific place or a region but the educational system worldwide [1–3,7]. This is one of the reasons the terminology of 'Emergency Remote Learning' (ERL) came to the surface. Hodges et al. (2020) [10] were among the first researchers to adopt the term 'Emergency Remote Learning' (ERL) in an attempt to distinguish it from typical online learning. More specifically, ERL refers to instruction that is being delivered in pressing conditions, such as COVID-19 or other natural disasters and emergency situations. The most significant difference between already established online courses and those converted to online in times of crisis is that the former are preceded by months of careful design and planning,

using a systematic model for their development [11], while the latter lacks the appropriate time for constructive development [10,12].

Certainly, assessment and evaluation practices in all their forms had to be aligned with the transformations related to emergency remote learning (ERL) due to COVID-19 [5,8,13,14]. However, the uncertainty of the pandemic in addition to the short timeframe available until the end of the academic semester left little time for the appropriate planning of *how* assessments should be implemented, especially during the first wave of the pandemic. Within this timeframe, these adaptations had to be effectively communicated to university instructors (the terms lecturer and instructor are used interchangeably) and students, who had no choice but to accept these new challenges. Moreover, instructors had to familiarize themselves with new assessment parameters, in addition to learning how to use the software necessary for these transitions. These were not easy tasks, especially in light of the technological challenges faced by higher education instructors, who varied significantly in their levels of preparedness in relation to their technology use and familiarity [2,9]. For example, a recent survey investigating the digital experiences of higher education instructors in the UK reported that only 34% of the teaching staff in higher education were given regular opportunities to develop their digital skills, despite the long-standing history of the UK in higher education and the offering of distance modules. Therefore, overall digital readiness was and still is considered as a major issue, exacerbated during the pandemic [3].

During the pandemic, institutes of higher education also had to face additional challenges related to 'Emergency Remote Assessment', which related to the technical [15] and security aspects of their examinations [13,16], while concurrently following the directives of their local or national quality assurance agencies. To start with, decisions had to be made regarding the format of the examinations [4,17]. Would take-home examinations be acceptable, or was it best to administer examinations through online platforms? In the cases where online platforms would be used, decisions had to be made regarding how the students would be invigilated while considering their privacy concerns [15]. Plagiarism and cheating were additional issues that concerned academics worldwide [15,18–20], especially in response to the remote emergency practices due to COVID-19 [21–24]. Some of the provisional reasons adhered to academic cheating proposed by Jose (2022) were lack of time, procrastination, lack of comprehension, lack of study habits, having a job in parallel to studying, low self-esteem, and the desire to obtain better marks [25]. Thus, instructors also had to familiarize themselves with new software related to plagiarism and cheating, in addition to having to adapt to new assessment types and online platforms in a very short period of time. In this nascent ecosystem, oral examinations were also considered as alternative assessment methods in the beginning of the pandemic [26,27]. However, the question remains—how did instructors evaluate these changes, and how effective did they consider them to be?

The purpose of this study was to critically examine the feedback obtained from higher education instructors regarding the implementation of the emergency remote assessment practices that took place within a private university in the Republic of Cyprus, in order to identify the strengths and weaknesses of the changes that took place. This was essential since the abruptness of the pandemic did not always allow for smooth transitions during the introduction of these changes. Therefore, it was imperative to collect data after the initial changes took place, to obtain feedback about them and feedforward for the semesters that followed. By critically examining these practices, institutions of Higher Education would be able to make more informed decisions regarding their online assessments, while providing suggestions on how they can be improved in the future.

Consequently, the research questions guiding this study were as follows:

1. To what extent did the lecturers feel comfortable with the changes in the examination process that took place during the COVID-19 pandemic, and what are the variables that can help explain their level of comfort?
2. What were the difficulties that were faced by the lecturers and their students due to the changes that were made to the examination process?

3. How did the lecturers evaluate the variations that might have occurred in student examination grades, and what do they attribute them to?
4. What are the lecturer's preferences regarding future examinations?

## 2. Materials and Methods

The data from this survey study were obtained from a web-based questionnaire that was sent out to all 205 lecturers who taught courses in an online MA program in Education in the Spring of 2020. This program, which is the largest within this university, was specifically selected, since it enabled the collection of larger amounts of data due to its many lecturers. Overall, 95 lecturers responded to the questionnaire, of which 54.7% were female. All lecturers had completed doctoral degrees, and had been teaching on average for 3.90 years, although that ranged from 1 year to 18 years. The majority of the lecturers that responded to the questionnaire were adjunct (86.3), with 13.7% being full-time faculty members.

The MA program that the lecturers taught in was offered in an online format through the Moodle platform. All the students in the program were from Greece or from the Republic of Cyprus. However, certain country regulations that were relevant to students from Greece, which composed the majority of the student population, required that they take face-to-face final examinations at the end of each semester for their degrees to be accredited. Since this was not feasible during the COVID-19 lockdowns, emergency adaptations had to be made to the testing process, which were eventually approved by the relevant accreditation agencies. Therefore, in May of 2020 an emergency decision had to be made at the administration level of the university on how to proceed with student final examinations. The decisions that were made regarding the examination process at the time were as follows:

1. The examinations process would take place from the students' homes through e-invigilation software (Proctorio) and would also enable the use of plagiarism identification software (Turnitin);
2. Additional time would be provided to the students to complete the examination in order to account for any technical problems that could possibly arise;
3. The examination would be supplemented with a short oral examination that would also take place online. Although these changes were designed to enable a smoother testing process, they also created some additional problems for certain students due to the technology requirements that were needed for the proper use of the e-invigilation software. For example, some students had older computers that did not meet the technology requirements needed, while others had faulty cameras and microphones which were essential for electronic invigilation. What made the issue even more complicated was the fact that supply-chain problems as well as the worldwide demand for computers and their relevant accessories made it difficult to purchase new computers, cameras, and microphones. As a result, this transitional period was not always smooth. With the completion of the examination process, more detailed information needed to be obtained to properly identify the issues that had occurred. This was essential to be able to identify any problematic aspects of the adaptations that had been made to the examination process, while concurrently utilizing this information to help guide the university through the following semesters. As a result, an online questionnaire was administered to all the lecturers of the specific program in June 2020, after the completion of all examinations.
4. The questionnaire that was administered through SurveyMonkey contained 25 closed response questions, which included background variables for the lecturers (gender, teaching role, years of teaching at the tertiary level), in addition to Likert-type questions asking them to evaluate their overall experiences regarding the adaptations that took place in the examination process. These questions revolved around the topics of (a) their opinions regarding usefulness of three new components that were added to the examinations which included the e-invigilation software, the extra test-

ing time, and the oral examinations, (b) the difficulties that they and their students faced, (c) their levels of comfort and satisfaction with the overall testing experience, (d) their evaluation of how grades differed from previous semesters with possible explanations, and I their preferences regarding any adaptations that may carry over to future semesters. Although most of the questions in the questionnaire were measured on a 5-point Likert scale, ranging from 1 ("Not at all"), to 5 ("A lot"), the first question that was administered, regarding the degree of comfort that they felt regarding the adapted examination process, was on a 4-point scale. This was purposefully decided in order to avoid the convenience of responding to the middle of the scale.

## 3. Results

Overall, the lecturers felt quite comfortable with this new examination process, since 24.21% responded that they felt extremely comfortable, and 53.68% felt very comfortable with it. Only 17.89% of the lecturers felt comfortable to a small extent, and 4.21% did not feel comfortable at all with this examination process.

A series of additional questions were asked within the questionnaire to identify the lecturers' overall opinions about the various aspects of the online examination. Therefore, they were first asked to evaluate the usefulness of three new components that were added to the examinations, namely the e-invigilation software, the extra testing time, and the oral examinations. Based on the results presented in Table 1, the lecturers tended to be overwhelmingly positive regarding the usefulness of the plagiarism detection software and the extra time that was available to the students to account for any technical difficulties they may have faced during the examination. For example, 41.49% found the plagiarism detection software to be very useful, and 25.53% found it extremely useful. Furthermore, 40.43% found the extra time available to be very useful, and 27.66% found it extremely useful. The results were not as strong regarding the oral exams, where the responses were more spread out within the five response categories, which ranged from 27.66% who evaluated the oral exams as extremely useful, to 12.77% who evaluated them as not useful at all. This pattern of results were further verified by the independent sample t-tests that were performed to determine whether the responses differed significantly from the midpoint of the scale. Of these results, only the usefulness of the oral examinations did not vary significantly from the midpoint of the scale ($t_{93}$ = 2.62, $p$ = 0.10), unlike the test adaptations of the plagiarism software and the extra time provided to the students.

**Table 1.** Lecturer evaluation of the usefulness of the changes made to the examination process.

|  | Plagiarism Software | Extra Time | Oral Examinations |
|---|---|---|---|
| Not at all useful (%) | 4.26 | 5.32 | 12.77 |
| To a small extent (%) | 8.51 | 10.64 | 15.96 |
| Useful (%) | 20.21 | 15.96 | 20.21 |
| Very useful (%) | 41.49 | 40.43 | 23.40 |
| Extremely useful (%) | 25.53 | 27.66 | 27.66 |
| Mean | 3.76 | 3.74 | 3.37 |
| SD | 1.06 | 1.14 | 1.38 |
| $t_{93}$ | 6.88 | 6.36 | 2.62 |
| $p$ | 0.001 | 0.001 | 0.10 |

When asked about whether their students faced any technical difficulties, 39.36% noted that their students did face technical difficulties. Additional questions were also asked regarding the lecturers' difficulties with this examination process. Table 2, below, describes the responses provided by the lecturers on these difficulties, by whether they

faced these issues, as well as the degree of how frequently they faced them. Overall, the majority of the lecturers did face difficulties with almost all of the problems mentioned in the questionnaire, although the degrees to which they were faced were quite small in most cases. For example, 75.53% of the lecturers indicated that their students faced camera or microphone problems, although their average response on the degree of the specific difficulty was 2.10 (SD = 0.90) on a 5-point scale. Furthermore, 70.21% of the lecturers indicated that they faced problems in trying to communicate with the students regarding the setup of the oral exams, although their average response on the degree of the specific difficulty equaled 2.38 (SD = 1.27). The next more frequently faced problems were those of dealing with plagiarism issues (61.29%), arranging dates and times for the oral exams (60.64%), and dealing with cheating (55.91%), although the average response for the cheating issues was only equal to 1.87 (SD = 0.99). Only 10.87% of the lecturers indicated that they faced problems with students being tested in the same location, thus, having the same IP address. On average, the lecturers faced (even to a small extent) 3.29 out of the 6 problems each (SD = 1.66), with only 8.4% of them indicating that they faced no difficulties in any of the problems listed in the questionnaire. These results are also demonstrated by the independent sample t-tests that were performed, where the average of the responses to all six problems all fell below the midpoint of the scale and were statistically significant at the 0.001 level.

**Table 2.** Difficulties faced during the examination process, as reported by course lecturers.

| | Cheating Issues | Plagiarism Issues | Camera or Microphone Problems | Students Testing in the Same Locations | Arranging Dates and Times for the Oral Exams | Trying to Communicate with Students about the Oral Exams |
|---|---|---|---|---|---|---|
| % of lecturers responding positively | 55.91 | 61.29 | 75.53 | 10.87 | 60.64 | 70.21 |
| No problems % | 44.09 | 38.71 | 24.47 | 89.13 | 39.36 | 29.79 |
| To a small extent % | 34.41 | 40.86 | 51.06 | 6.52 | 26.6 | 34.04 |
| Some problems% | 13.98 | 9.68 | 17.02 | 1.09 | 12.77 | 10.64 |
| Many problems% | 5.38 | 8.6 | 5.32 | 2.17 | 10.64 | 19.15 |
| Very many problems % | 2.15 | 2.15 | 2.13 | 1.09 | 10.64 | 6.38 |
| M | 1.87 | 1.95 | 2.1 | 1.2 | 2.27 | 2.38 |
| SD | 0.99 | 1.01 | 0.9 | 0.67 | 1.36 | 1.27 |
| T | −10.98 (df = 92) | −10.02 (df = 92) | −9.69 (df = 93) | −25.95 (df = 91) | −5.23 (df = 93) | −4.71 (df = 93) |
| P | 0.001 | 0.001 | 0.001 | 0.001 | 0.001 | 0.001 |

The lecturers were also asked to compare their students' examination grades during this examination period, compared to the examination grades of previous semesters. Exactly 50.00% of them indicated that their grades were about the same in both semesters. However, 28.26% of them indicated that the student grades were lower this semester, compared to 21.74% who indicated that they were higher. To try to provide possible explanations for these results, the lecturers were asked to indicate their degree of agreement with certain hypotheses for the variations in the grades that they might have observed. As such, the lecturers who indicated that their students' grades were higher during COVID-19 responded to some possible hypotheses due to which the higher grades were observed (*n* = 26), while the lecturers who indicated that their students' grades were lower during COVID-19 responded to some possible hypotheses due to which the lower grades were observed (*n* = 20). The results presented in Table 3 do not provide any clear patterns, since the mean responses for almost all possible explanations fell within the middle of

the five-point scale. The only result that was clearly above the midpoint of the scale was the hypothesis that the students received higher grades because they had more time to complete the test (M = 3.38, SD = 1.07).

**Table 3.** Possible explanations for the differences that were observed in examination grades this semester.

| | Not at All | To a Small Degree | To Some Degree | To a Large Degree | To a Very Large Degree | M | SD |
|---|---|---|---|---|---|---|---|
| **Higher grade explanations (*n* = 26)** | | | | | | | |
| The students had more opportunities to cheat | 33.33 | 38.10 | 14.29 | 9.52 | 4.76 | 2.14 | 1.15 |
| The students had more time to complete the test | 0.00 | 23.81 | 33.33 | 23.81 | 19.05 | 3.38 | 1.07 |
| The test was easier | 33.33 | 38.10 | 19.05 | 0.00 | 9.52 | 2.14 | 1.20 |
| **Lower grade explanations (*n* = 20)** | | | | | | | |
| The students had more difficulties in studying for the test due to COVID-19 | 17.24 | 51.72 | 31.03 | 0.00 | 0.00 | 2.14 | 0.69 |
| The students did not have enough time to complete the test | 26.67 | 33.33 | 26.67 | 10.00 | 3.33 | 2.30 | 1.09 |
| The test was more difficult | 26.67 | 26.67 | 20.00 | 16.67 | 10.00 | 2.57 | 1.33 |

For obtaining an even more concrete picture of the lecturers' views of this examination period, a regression was run in SPSS 28 to explain the levels of comfort that they had in relation to the examination process. The analysis was statistically significant ($F_{7,81}$ = 11.34, $p$ = 0.001), and it explained 49.49% of the variation of the dependent variable. The results of the analysis presented in Table 4 shows that the variables that were statistically significant were the usefulness of the software for locating plagiarism ($\beta$ = 0.17, $p$ = 0.010), the usefulness of the oral exam ($\beta$ = 0.22, $p$ = 0.001), as well as the number of problems that the lecturers had to face throughout the examination process ($\beta$ = −0.97, $p$ = 0.014). The lecturers' background variables, such as their gender and the years they have been teaching in the program, were not statistically significant in this analysis.

**Table 4.** Regression results of lecturer satisfaction with the new examination processes.

| | Unstandardized Coefficients | | Standardized Coefficients | t | p |
|---|---|---|---|---|---|
| | B | Std. Error | Beta | | |
| (Constant) | 2.11 | 0.40 | | 5.33 | 0.010 |
| Years teaching | −0.01 | 0.02 | −0.06 | −0.65 | 0.521 |
| Gender (2 = female) | −0.12 | 0.12 | −0.08 | −0.98 | 0.329 |
| Student technical problems | −0.23 | 0.13 | −0.15 | −1.82 | 0.073 |
| Plagiarism software | 0.17 | 0.06 | 0.24 | 2.65 | 0.010 |
| Extra time | 0.05 | 0.06 | 0.07 | 0.76 | 0.450 |
| Oral examination | 0.22 | 0.05 | 0.41 | 4.55 | 0.001 |
| Number of problems faced | −0.97 | 0.04 | −0.21 | −2.51 | 0.014 |

Finally, the lecturers were also asked to identify their preferences regarding the examination process for the following semester. What is overwhelmingly evident from Table 5, below, is that there is a strong preference for the face-to-face examination format by the large majority of the lecturers (M = 4.13, SD = 1.12), followed by examinations through electronically invigilated software, where the invigilation is performed automatically by the software itself (M = 3.75, SD = 1.04). This was followed by take-home examinations

(M = 3.23, SD = 1.40), and then by remotely invigilated examinations, where the invigilation is performed live by the lecturers themselves (M = 2.86, SD = 1.24). When asked to choose only a single way of invigilating examinations in case there were COVID-19 measures the following semester as well, 52.13% of the lecturers preferred the option of electronically invigilated examinations by software; 38.30% preferred a take-home examination, while only 9.57% preferred the option of remotely invigilated examinations by the lecturers themselves. The independent sample t-tests that were performed (based on the midpoint of 3) were only statistically significant at the 0.001 level for the face-to-face examinations as well as for the electronically invigilated examinations, which were the two formats with which the lecturers appeared to be the most comfortable.

**Table 5.** Preferences regarding the degree of comfort regarding examination types for future semesters.

| | Face-to-Face Examination | Electronically Invigilated Examination | Remotely Invigilated Examinations (by Lecturers) | Take-Home Exam (No Invigilation) |
|---|---|---|---|---|
| Not at all comfortable | 4.26 | 4.21 | 13.68 | 14.74 |
| To a small extent % | 7.45 | 7.37 | 29.47 | 18.95 |
| Comfortable % | 8.51 | 22.11 | 27.37 | 18.95 |
| Very comfortable % | 30.85 | 42.11 | 15.79 | 23.16 |
| Extremely comfortable % | 48.94 | 24.21 | 13.68 | 24.21 |
| M | 4.13 | 3.75 | 2.86 | 3.23 |
| SD | 1.12 | 1.04 | 1.24 | 1.40 |
| t | 9.77 (df = 93) | 7.00 (df = 94) | −1.07 (df = 94) | 1.62 (df = 94) |

## 4. Discussion

Due to COVID-19, multiple decisions and a variety of adaptations had to be made, especially during the first wave of the pandemic, in many aspects of higher education including assessment procedures, practices, and tools. However, the abruptness with which these changes occurred did not allow for adequate research to be performed on them prior to their official implementation. As a result, it was important to critically examine the feedback obtained from higher education instructors regarding the implementation of the emergency remote assessment practices in order to identify their strengths and weaknesses. By critically examining these practices, institutions of higher education can make more informed decisions regarding their online assessments, while providing suggestions on how they can be implemented in the future. The feedback that was obtained from the current study revolved around four aspects of assessment, as follows: (1) the usefulness of three new components added to the examination process, (2) the technical and plagiarism issues that they might have faced, (3) the differences in examination grades between face-to-face and emergency remote assessment adaptations, and (4) the instructor's overall satisfaction and evaluation of the new examination processes and their specific preferences.

Overall, the course instructors responded positively that all three adaptations that were made in the assessment process of their courses were useful to a large extent. More specifically, it is not surprising that they considered the use of plagiarism software as the most useful addition in the e-assessment process. This result is most likely a reflection of their concerns regarding cheating and plagiarism [15,18–20]. Although academic and assessment dishonesty are not something new, it seems the COVID-19 pandemic has brought new weight to these concerns as well as new aspects to them. This is not unexpected, since studies have found that online assessments, and especially the ones in which the students feel more "distant" from their instructors such as in online courses, are more likely to have students who cheat in numerous ways [26,28]. Similar concerns with online

assessments were identified by faculty members in a study by Meccawy, Meccawy, and Alsobhi (2021) [29], who indicated that the ways of preventing cheating and plagiarism were the main challenges that they faced regarding student assessment. The same study also indicated that cheating and plagiarism were both issues that had increased considerably during the pandemic [29]. Although lecturers have been trained to some extent in emergency remote assessment tools and are making tremendous efforts to overcome student cheating, the extent of control remains questionable.

TIe instructors that took pIrt in this study also responded positively to the usefulness of providing additional time to the students during the examination process in an attempt to compensate for any technical or other issues adhered during their examinations. This is similar to the practice of providing additional testing time to students with learning disabilities [30]. As a practice in relation to COVID-19, however, this is controversial. Although, on the one hand, researchers, such as Jose (2022), claim that the absence of enough time may cause students to cheat in an attempt to cope with the testing situation, it is also possible that having too much time to respond to a test might also lead to unethical behaviors on the part of some students. Therefore, providing the "ideal" amount of time to students during online assessments in order to be fair to the students, while discouraging unethical behavior, is an issue that needs to be looked into in more detail in the future [31].

The sample participants were a bit less positive regarding the oral examinations that they had to perform after the students' written examinations. The initial goal of this oral examination was to enable the instructors to ask any clarifying questions to the examinees. These questions could either revolve around the content of their responses on the written test, or around issues related to cheating and plagiarism attempts. At the same time, this process enabled the examinees to unofficially provide any comments to their instructors regarding problems that they might have faced while taking their tests. One possible hypothesis for why the instructors were a bit less positive regarding the oral assessment is that this new component that was added to the assessment adaptations was very subjective and open-ended, which made it difficult for the instructors to objectively utilize it without harming the validity of the student examination scores. This is amplified by the lack of scientific literature on the overall topic of oral examinations in online environments [26] either before or after the pandemic.

Not surprisingly, the instructors indicated that a lot of problems were faced by both their students and themselves throughout the emergency remote assessment process. These problems included cheating, plagiarism, having students tested from the same location, camera and microphone problems, as well as communication problems with the students while trying to set up their oral examinations. All these issues were unique from the instructors' experiences in previous semesters and were problems that were not relevant in previous testing situations. Moreover, although some instructors only ended up facing some of these problems, other instructors actually faced all of the issues mentioned above. Since these were problems that were not typically faced in the past, there were no quick answers on how to resolve them, and the instructors had to spend a lot of additional time on these matters. This was further amplified by the fact that the pandemic did not allow for adequate time to train the instructors and their students on emergency remote assessment solutions (e.g., e-proctoring exams) [32].

All these issues were bound to have an impact on the instructor's perspectives on the assessment process. A regression analysis that was performed revealed that the instructors who were most satisfied with the assessment process were the ones who had students facing less technological problems, and who themselves had faced the least number of problems during the examination. The regression analysis has also shown that the most satisfied lecturers where the ones who were also most satisfied with the use of the plagiarism software and the oral examination.

Based on all these aspects of the assessment adaptation, the instructors were asked to state their preferences regarding which form of assessment they would prefer for future semesters. What was especially interesting was the fact that the overwhelming majority of

instructors preferred face-to-face examinations. This might be due to a number of reasons. First, this might be a reflection of their concerns related to cheating and plagiarism. Although most testing situations have these issues which can never be completely eliminated, they are much more prominent in online assessments [29,32]. Therefore, these instructors might believe that they might be much easier to handle in face-to-face examinations. Another possible explanation for this result might be due to the fact that these assessment procedures were implemented so quickly, that the instructors did not have adequate time to properly evaluate their various aspects with their advantages and disadvantages. To overcome these problems in the future, it is important to make an effort to provide adequate guidelines to instructors on the proper use of these aspects of assessments, in order to ensure that they are implemented properly, while having a clear understanding of their purpose. This is in accordance with Montenegro-Rueda, Luque-de la Rosa, Sanchez-Serrano, and Fernandez-Carero (2021) [32], who also highlight the need to improve the training of instructors regarding e-assessments in online platforms.

As with all studies, the current study is not without limitations. A major constraint of the study is the fact that the data collected originated from a single university for one degree program. As a result, only the adaptations that were made within this university were evaluated. Many more adaptations have evolved in other countries and universities, which were not, however, included in the current study. Moreover, all the study participants had already experienced online learning and online platforms, since they were teaching in an online program and, thus, had a precedent of teaching and assessing in such an environment. As a result, their opinions regarding online assessments might not be comparable to instructors who had never taught remotely before the pandemic. The issues faced by the students (as evaluated by their instructors) also might not be comparable to those of students who had not been enrolled in online programs, since the current students were already required to have some technological experience for their studies. Therefore, the generalizability of the study is limited within this framework.

An important component that was missing from the current study regarded the students' opinions on the emergency remote assessment practices adopted by their university. Therefore, further studies are recommended to examine this issue from a student perspective. It would also be interesting to examine whether the student experiences and opinions varied based on whether they were attending online or face-to-face programs, based on their field of study (e.g., whether students in technologically based majors differ from students in education), their age, background characteristics, as well as based on their degree level (e.g., bachelor vs. masters).

This study did not provide any clear consensus regarding the instructor's perspectives on whether student grades increased or decreased during the first semester of the pandemic. There was also no clear consensus regarding the reasons hypothesized by the instructors on why such changes might have occurred. However, instructor perspectives on such issues cannot always be considered as reliable, especially when their examinations are not standardized and equated properly. What would be more important and more reliable though, would be to actually compare student grades from before and after March of 2020 to be able to obtain more reliable and valid results on these issues. Further studies should take this into consideration as well.

Finally, since we are now entering the post-pandemic era and the era of ChatGPT, it is important to follow up on university assessment and e-assessment practices. By this timepoint, six semesters have passed since the first COVID-19 lockdowns, which has provided adequate time for universities to reconceptualize their e-learning as well as their e-assessment practices. Therefore, since different types of assessment adaptations have been adopted based on each country's and each university's individual characteristics, it is important for future studies to examine these variations and their impact, in order to gain further insights into these practices and to enable an improved testing experience that helps obtain test scores that are valid and reliable to the maximum extent possible.

**Author Contributions:** The conceptualization of this study, the collection of the data and the data analysis was performed by E.C.P. G.S. was mostly involved in the writeup of the introduction and the literature review. Both authors were involved in the writeup, review, and editing of the manuscript. All authors have read and agreed to the published version of the manuscript.

**Funding:** This research received no external funding.

**Institutional Review Board Statement:** The study was conducted in accordance with the Declaration of Helsinki, and approved by the Research and Ethics Committee of the School of Education Sciences of University of Nicosia (protocol code 2020/5/3 and date of approval 30 May 2020).

**Informed Consent Statement:** Informed consent was obtained from all subjects involved in the study.

**Data Availability Statement:** The data can be obtained from the authors upon request.

**Conflicts of Interest:** The authors declare no conflict of interest.

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
