# Peer review of "Evaluating Emergency Remote Assessment Adaptations in Higher Education due to COVID-19: Faculty Insights and Challenges"

_education, doi:10.3390/educsci13020184_

Round 1

Reviewer 1 Report

The method can be improved by providing more detailed information about the type of question in the questionnaire. 

Author Response

Thank you very much for the very thoughtful comments. The abstract is now updated, clarifying the method used as well as the sample.

Regarding comment in Section 2,  line 130 onward, the knowledge of dated technology equipment of students, comes from the authors' experience and background knowledge as teachers of the modules, thus personal complaints and email from the students themselves and not from specific data percentage sample retrieved for this paper.  These were mere observations based on which we decided to create this questionnaire and gather more reliable data.

Regarding comment in Section 3, line 158, we went ahead and included more detailed information on the questions that were included in the questionnaire in the methods section. 

Reviewer 2 Report

The identified research problem is currently significant. For the study, adequate recent literature has been referred to. The research method is also appropriate. However, the following issues should be addressed while revising the paper.

1.     It is preferable if you can improve your writing in the abstract, such as a brief introduction, research problem, objectives, methods, findings, and recommendations.

2.     There is no mention of the population or the sample. The MA program's lecturers have all been chosen for the study. It should be stated how many lecturers are associated with that program and how many work as invigilators/instructors. Also, as you have used both terms 'lecturers' and 'instructors' throughout the paper, it should be clear whether the lecturers are working as instructors or others. The e-invigilation software used is not clearly introduced. Please revise.

3.     The study takes the features of a case study, and then the author should provide a separate section about the selected Degree program while specifying it in the title (optional).

4.     Need a strong justification to select that particular Degree program and selecting all attached to the e-assessment process.

5.     There must be a compelling reason to select that particular Degree program and all associated with the e-assessment process.

6.     The paper contains some grammatical and formatting errors (some were marked in the paper). Please proofread them thoroughly.

7.      Double-check the referencing style.

8.      Reviewed paper is attached herewith.

Author Response

Thank you for your very thoughtful comments. We went ahead and made all corrections that were suggested regarding the use of the English language.

Regarding comment 1, the abstract now includes the information that was requested. 

Regarding comment 2, we have also added more information about the actual population number. We also created a footnote to clarify that we use the terms instructor and lecturer interchangeably in this manuscript. We have also included the name of the e-invigilation software that was used, along with the name of the plagiarism software.

Regarding comments 3-5, we have clarified the reason why this specific program of study was selected for this study, which was basically due to its large size, which enabled us to collect more data in order to get more reliable responses.

The references have also been reorganized appropriately and corrected based on the relevant changes that were made.

Reviewer 3 Report

Dear authors, I have some important comments and revision suggestions for you. I hope you consider them. 

Where is Cyprus? North or South? Which one? Cyprus is not a country! It is the Greek Administration of Southern Cyprus. Should revise it.

What is your research method? I could not see it. You should add it and should explain it.

The method section is unclear. What are your sampling selection and analysis methods? Should explain it.

What is your research paradigm? Qualitative or quantitative? Should explain it.

You use mostly frequencies and percentages. These are beginner analysis methods. You should use more complex tests.

You should use more references for related studies and discussions. You consider geographical diversity. 

Author Response

Thank you for your thoughtful comments. We have clarified that the data have been collected from the Republic of Cyprus so that there is no misinterpretation in any way . We have also clarified that this was a survey study, and that the questionnaires were sent out to all members of the population (which is now referenced more clearly). We also proceeded with including inferential statistics in this study as well, as requested. The references have also been updated to be organized accordingly based on the journal requirements. 

Round 2

Reviewer 3 Report

Dear authors, thanks for your great efforts in revising the manuscript. Congratulations!